# Relationship between outpatients' sociodemographic and belief characteristics and their healthcare-seeking behavioral decision-making: Evidence from Jiaxing city, China

**Mingming Yu** [1]*, **Zan Yang**[2], **Cheng Jiang**[1], **Lemin Shi**[3]

1 Department of Economics and Management, Tongji Zhejiang College, Jiaxing, Zhejiang, China, 2 Faculty of Science, Tongji Zhejiang College, Jiaxing, Zhejiang, China, 3 Health Education Center, Jiaxing Municipal Health Commission, Jiaxing, Zhejiang, China

* yigemeng_1@163.com

## Abstract

### Background

China established the Tiered-network Healthcare Delivery System (THDS) in 2015 to address the disproportionate number of patients attending tertiary hospitals relative to primary- or secondary-care institutions. Although the reported number of outpatients visiting tertiary hospitals is slowly decreasing, numerous patients choose to visit them regardless of their disease's severity. To effectively implement the THDS, this article explored the relationship between patients' sociodemographic and belief characteristics and their healthcare-seeking behavioral decision-making in China.

### Methods

Data obtained through questionnaires were analyzed using decision tree and logistic regression models to explore outpatients' characteristics and medical decision-making using comprehensive feature data. Moreover, further statistical analyses were conducted on the outpatient data obtained from the regional population health platform in Jiaxing, China.

### Results

The decision tree model revealed that whether outpatients have medical insurance is the primary factor guiding their healthcare-seeking behaviors, with those without medical insurance more likely to choose primary or secondary hospitals to treat minor diseases. For those with medical insurance, profession is the main factor, with industrial workers more inclined to choose primary or secondary hospitals for minor diseases. The logistic regression analyses revealed that outpatients without insurance and who were not freelancers or individual owners were more likely to choose primary or secondary hospitals for minor diseases. Further statistical analysis of the data from the Jiaxing population health platform showed that, for minor or general diseases, outpatients without medical insurance and

**Data Availability Statement:** All relevant data are within the paper and its Supporting Information files.

**Funding:** This research was funded by "2018 Jiaxing first batch of science and technology plan "(public welfare research plan), grant number "2018AY32039." The funders had no role in study design, data collection and analysis, decision to publish, or preparation of the manuscript.

employed as farmers tended to choose primary and secondary hospitals over tertiary hospitals.

## Conclusion

The three analyses yielded consistent results: in China, medical insurance and patients' profession are the most important factors guiding outpatients' healthcare-seeking behaviors. Accordingly, we propose that the government should focus on economic reforms to increase outpatients' visits to primary and secondary hospitals and diagnosis-related groups' payment of medical insurance to decrease the admittance of patients with minor diseases in large tertiary hospitals. Meanwhile, the government should correct patients' belief prejudice about selecting hospitals through corresponding publicity.

## Introduction

China's current medical system is the product of its socialist and managed capitalist economic system [1]. Public and private hospitals have coexisted, with the former being the majority (by the end of 2019, public hospitals' beds accounted for 72.5% of the total number [2]). As per the national medical service system, public hospitals are classified as primary, secondary, or tertiary based on their size and the level of healthcare provided [1]. For instance, tertiary hospitals provide the highest level of treatment and specialization. These institutions have received consistent policy attention and support through financial subsidies [3], talented personnel, and loans, as well as made full use of the market economy system reforms in the past 30 years. Moreover, tertiary hospitals have retained considerable revenue by adding a 15% mark-up on drug sales and setting high prices on high-tech diagnostic tests [4,5]. The 15% mark-up policy permitted by the government on drug prices allows public hospitals to retain 15% profits of drug prices after selling them to patients. Hence, public hospitals could be self-sufficient at a large cost, and financial subsidies account for only a small part of their income.

Although this situation has been gradually changing since China's 2009 Health Policy Reform [4,5], which includes a zero-mark-up drug policy and reduced prices for diagnostic tests, tertiary hospitals have created a monopoly advantage. Thus, many patients choose to be treated in tertiary hospitals, regardless of the diseases' severity [3]. By the end of 2019, 23.6% of Chinese outpatients and 39.4% of inpatients were concentrated in tertiary hospitals, which only account for 0.27% of all hospitals, despite the average outpatient and inpatient cost ratio in tertiary, secondary, and primary hospitals being 3:2:1 (337.6:214.5:142.6, Unit: RMB) and 4:2:1 (13.7:6.2:3.3, Unit: Thousand RMB) [2], respectively.

Consequently, tertiary hospitals are often overcrowded and Chinese patients increasingly experience difficulties in and high costs of obtaining medical services. To address this, in 2015, China's administration established the Tiered-network Healthcare Delivery System (THDS) [6]. In adopting THDS, patients are encouraged to go to primary hospitals first, where those with severe disease will be referred to secondary or tertiary hospitals, and they will return to primary hospitals for rehabilitation when they are in stable condition [6]. However, the THDS is difficult to implement because of the liberalization and marketization of patients' choices of medical treatment and the small price gaps between primary, secondary, and tertiary hospitals [7,8]. In 2013, a survey of six cities in China revealed that 62.2% of patients felt uncomfortable choosing primary care institutions for their first medical visit [9], however, the number of patients visiting these institutions is slowly increasing [10]. Since the THDS was not implemented satisfactorily, China has strengthened the THDS reform since 2017 [11]. This included

price modifications, which led to significant reductions (until 2019, it has fallen to 23.1% after 2017) in the number of outpatient visits to tertiary hospitals in Beijing city [12].

The reduced number of outpatients visiting tertiary care institutions following the 2017 THDS reform suggests the following question: What type of patients would choose to visit primary and secondary hospitals instead of tertiary institutions? Answering this is tantamount to drawing a "portrait" of the patient by considering comprehensive features, which, if undertaken correctly, could improve public medical administrators' understanding of patients' medical decision-making. This, in turn, could facilitate the efficient equitable distribution of medical resources and effective guidance of patients consistent with standards of care.

The abovementioned portrait would reflect the relationship between the patients' sociodemographic and belief characteristics and their healthcare-seeking behavioral decision-making. While this relationship has been analyzed in previous studies regarding specific diseases or populations [13–15], it has not been studied regarding the effective implementation of the THDS. Moreover, although there is evidence from existing research that uses multivariable logistic regression and other statistical tests to analyze the factors that affect patients' behavioral intentions [16–18], there is minimal information on using technology (e.g., machine learning classifier) to examine the relationship between the variables. To fill these gaps, we conducted a comprehensive empirical analysis of healthcare-seeking behavioral decision-making based on a theoretical framework.

## Materials and methods

### Theoretical foundation

The theoretical framework used in this research was built by combining Andersen's behavioral model of health services use and prospect theory. The model involves systematic research, including population features, such as 1) Predisposing factors that can be categorized into personality factors (age, gender, and other demographic factors), social structure (education, profession, race, and social network among others), and health beliefs (people's attitudes, values, and knowledge of health and medical services). These affect people's perception of the use and demand intensity of medical services. 2) Enabling factors that promote or hinder people's ability to use medical services, including personal, family, and community resources. Personal resources include income, medical insurance, and time to arrive at medical institutions among others. 3) Perceived personal medical service needs that are affected by social factors and health beliefs and reflect how people view their health and functional status, experience disease symptoms and pain, and worry about their health [19].

Prospect theory is based on the individual's state of decision-making. According to it, subjective systematic bias and individual risk preference directly affect the decision-making process and the patient's interpretation and value judgment of the expected goal, thus affecting their healthcare-seeking behavioral decision-making. Subjective systematic bias varies with cognitive ability and disease severity [20].

According to the theoretical framework and several important factors influencing healthcare-seeking behavioral decision-making that have been generally recognized by scholars in recent years, such as economics, distance from hospitals, and self-report of the disease [21–23], we selected seven patient characteristics that comprehensively reflect sociodemographic and belief features based on the research paradigms of behavioral theories (See section 1 in S1 File for more details).

### Survey and questionnaire design

Zhang LL and other scholars [24] attributed the need for regional medical services, which is manifested in outpatients' healthcare-seeking behaviors, to demographic and economic

factors. Moreover, surveying outpatient departments instead of specific communities and units better reflects the randomness and reality of a city. Therefore, we administered our survey and questionnaire to patients presenting at the outpatient clinics in Jiaxing's several hospitals, considering Jiaxing as a representative, medium-sized city in China.

The questionnaire comprised two parts. The first part collected data about the seven representing sociodemographic and belief characteristics of the outpatients. The seven patient characteristics were selected according to the theoretical foundation described earlier, which were gender (male or female), age (<40, children or youth; 40–59, middle age; >59, old age), profession, birthplace (in Jiaxing, near Jiaxing, or distant from Jiaxing), residence area (in Jiaxing, near Jiaxing), having medical insurance (yes or no), and self-report of their disease's severity (serious, medium, or minor). The options after all characteristic variables were chosen by the survey subjects according to their situation. "In Jiaxing" or "near Jiaxing", includes the two districts, two counties, and three county-level cities. Some patients were working and living in the central urban area, that is, the urban districts, and were characterized as living "in Jiaxing" in our questionnaire survey, while those characterized as living "near Jiaxing" generally worked and lived in the surrounding counties and county-level cities. "Distant from Jiaxing" denotes cities other than Jiaxing. Outpatients living "near Jiaxing" generally use buses and cars to reach the hospitals in the central urban area, and experience longer transportation time (usually 30 minutes to one hour by car, and longer by bus). Therefore, compared with outpatients living in the central urban area, who face shorter transportation time, (usually less than 30 minutes by car), those living in the surrounding counties and cities were inconveniently far.

Financial status is important in our research. However, to respect patients' privacy, their profession was used as a proxy for income (e.g., farmer, industrial worker, staff or civil servant, freelancer or individual owner), given that, in China, they are correlated to some extent [25].

The second part of the questionnaire addressed patients' healthcare-seeking behavioral decision-making regarding the THDS by the following question. The respondents were asked about their general behaviors when seeing a doctor; specifically, whether they attended a large hospital (In China, residents generally do not pay attention to the official classification of hospitals. Generally speaking, the large hospitals in their minds are often tertiary hospitals, and the small hospitals are primary and secondary hospitals.) regardless of the disease's severity (option A), or whether they visited a large or small hospital for serious or minor disease, respectively (option C).

The sample size was 200 patients, calculated using the following formula: $N = Z^2 \times (P \times (1 - P))/E^2 = 196$ (1.96×1.96×0.5×0.5/0.07×0.07 = 196), where Z score is 1.96 when the degree of confidence is 95%, and P is the probability value, generally 0.5, and E is the sampling error rate taken as 7% [26]. Regarding the selection of samples, we adopt the method of stratified sampling, so the questionnaire was administered in the outpatient departments of five tertiary and secondary public hospitals and one public primary hospital in Jiaxing. The former included Jiaxing First Peoples Hospital, Jiaxing Traditional Chinese Medicine Hospital, Jiaxing Second Peoples Hospital, Jiaxing Armed Police Hospital, and Jiaxing Third Peoples Hospital. Their outpatient visits in 2017 separately accounted for 24.7%, 23.2%, 21.2%, 5.6%, and 5% of the total number of outpatient visits in Jiaxing, respectively, for a total of 79.7% of all outpatient visits. In contrast, visits to primary hospitals in Jiaxing accounted for 15.2%, while visits to other secondary and township hospitals accounted for 5.1% [27]. Based on the proportion of outpatient visits in the city, we distributed the 200 respondents as follows: 168 respondents across the five tertiary and secondary hospitals and 32 to Nanhu Community Health Center (a primary hospital, we surveyed this hospital to represent all primary hospitals in Jiaxing), with the ratio of respondents being 5.3:1 (roughly in line with the ratio of 79.7%:15.2% = 5.2:1). Given the small proportion of "other" hospitals (i.e., other secondary or township hospitals), we did not include them in our research (See section 4 in S1 File for more details).

To ensure that the sample was randomly selected under the same condition, in given hospitals, all questionnaires were administered simultaneously (July 5, 2018) and the respondents were selected using random samples based on outpatient ID numbers, without excluding any disease: From the administrative patient admission system of the outpatient clinics, we selected the outpatients' ID by random drawing of lots in Excel. The selected patients were instructed to complete the questionnaires based on their general health-seeking behaviors before the treatment. Only three and two outpatients in the tertiary and primary hospitals, respectively, refused to participate. Thus, 195 complete questionnaires were obtained. The survey process was approved and supervised by the Municipal Health Commission of Jiaxing, and all participants signed written informed consent forms before completing the questionnaire.

## Decision tree analysis

The outpatients' responses regarding their general healthcare-seeking behaviors were classified into groups using a C4.5 classifier to estimate a decision tree model: This type of classifier is among the most commonly used in classifying patterns of machine learning [28]. Moreover, machine learning is concerned with features and labels. Features are observable attributes of the observed object, and labels are what we want to predict. In this research, the main decision tree task was to extract useful features and construct the mapping from features to labels; in other words, to draw a portrait.

By Python's sklearn package, the PyCharm (V 2019.3.1) [29] implementation gave the information gain rates of C4.5 for gender and self-report of the disease's severity were c. 0.1% and smaller; the contributions were significantly small due to cross-entropy loss. Additionally, from a common-sense perspective, self-report of the disease's severity (this time to see doctor) is unrelated to the questionnaire (i.e., the patient's general behaviors). Therefore, as the model has other features and more samples are needed, we conducted another decision tree model without these two features and reanalyzed it using the C4.5 classifier. Moreover, we applied the holdout method to solve the overfitting problem, which used estimation and validation steps. The pruning rule for the validation steps used was Cost Complexity Pruning (post-pruning) [30]. After the 13th prune, the minimum error proportion in the validation set was 20.5%; accordingly, we chose this as the final decision tree model. Finally, the model was drawn using Matplotlib of PyCharm (V 2019.3.1) (Fig 1) (See section 5 in S1 File for more details).

## Logistic regression analysis

Decision tree and logistic regression are two different classification methods. In many studies, with the same datasets, the two classifiers were compared by the sensitivity or specificity to the predictive performance in machine learning [28,31]. Lee Yoonju [32] also got the common major predictors of suicide attempts in the two methods, showing the robustness of results using the two methods. Hence, we conducted the logistic regression analysis to compare and advance the results of the decision tree, with the datasets of questionnaires.

The dependent variable comprised participants' responses to whether they attended a large hospital regardless of the disease's severity (option A, which was coded as 0) or attended a large hospital for serious diseases and a small hospital for minor diseases (option C, which was coded as 1). The independent variables were the outpatients' sociodemographic and belief characteristics, which were dummy-coded. The model was specified as follows: $logit(p_i) = \varphi + \alpha N_i + \beta O_i + \gamma P_i + \delta Q_i + \varepsilon R_i + \iota S_i + \theta T_i$, where, N is gender, O is age, P is profession, Q is birthplace, R is residency location, S is having or not having medical insurance, and T is the self-report of the disease's severity. The remaining letters and i represents coefficients and

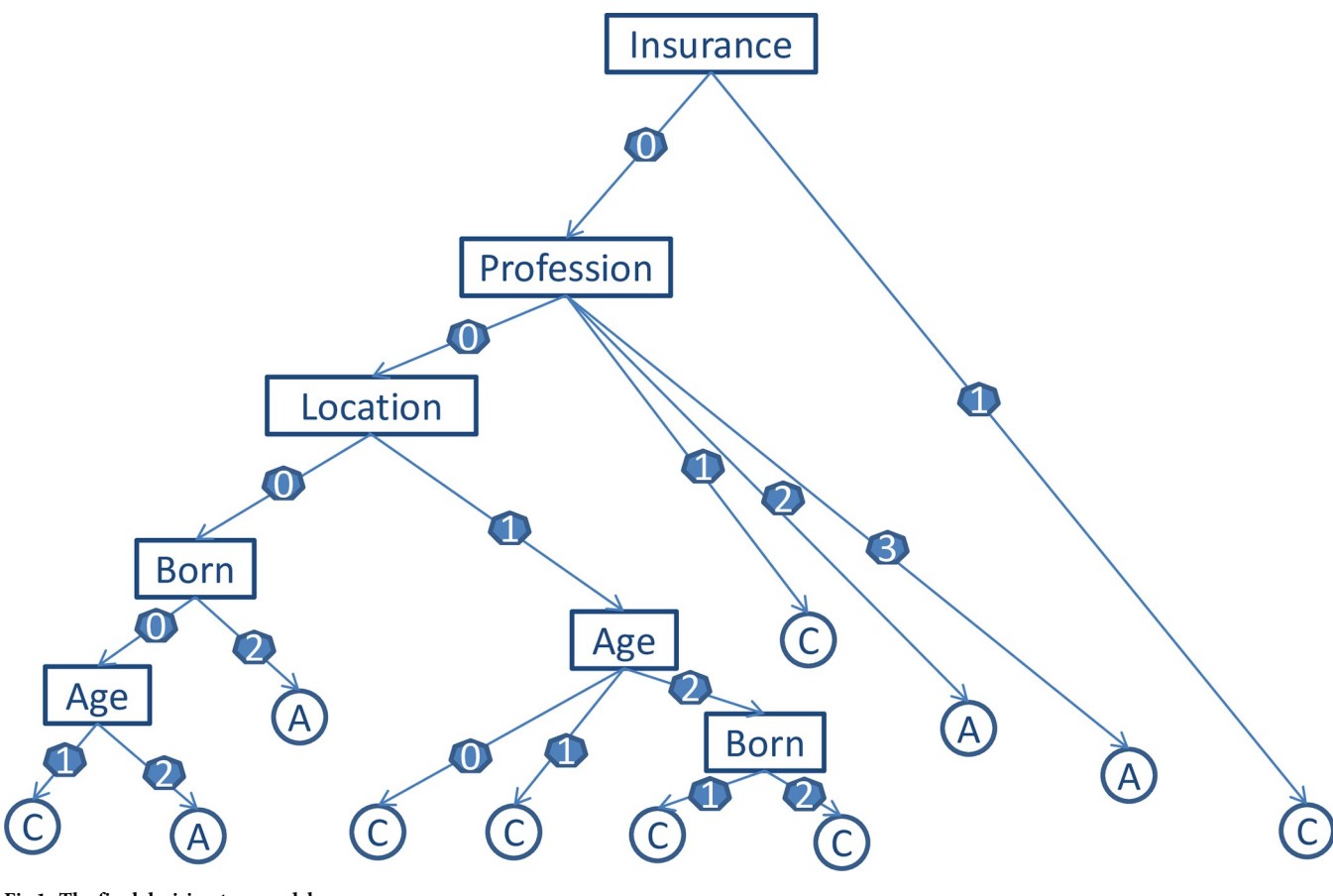

**Fig 1. The final decision tree model.**

outpatients' order, respectively. Backward-stepwise regression analysis was used with the variables with p value < .05. The data were analyzed using Stata (V13.1) [33].

## Analysis of the data from the population health platform

The previous two analyses used the questionnaire survey data were based on the decision-making of outpatients' behavior. The survey was limited by the possible inconsistency of patients' external expression and real behavioral consciousness, and the hospitals of survey in the city were representative but not comprehensive. Moreover, if different data and methods can get similar results, it shows that they are robust. Hence, to verify our conclusions from the questionnaires were reliable, we further analyzed additional data available through Jiaxing's population health platform of outpatients who had visited the city's hospitals. We randomly selected two sample datasets comprising 12,464 outpatients (dataset 1) that visited tertiary and 10,237 outpatients (dataset 2) who visited the city's secondary or primary hospitals from 2017 to 2019, respectively. These patients were suffering from ailments such as cold, hypertension, diabetes, and bronchitis. Neither were the classification and staging of severe or middle and late stage reflected in the diagnostic names nor were there any complications and associated diseases. We considered these diseases "minor or general" because, they were excluded from the scope of the serious disease insurance scheme specified by the State [34], and we focused on outpatients' social belief levels, rather than scientific clinical regulation (See section 3 in S1 File for more details).

We compared the proportion of outpatients with and without medical insurance as well as the proportion of farmers and non-farmers across the two datasets (we could only obtain information about professions regarding farmers or non-farmers) using Pearson's chi-square test, and the differences were statistically significant (p value < .05). Moreover, we compared the results from Jiaxing's population health platform and the decision tree model regarding the relationship between outpatients' characteristics and their compliance with the THDS. The data were analyzed using Stata (V13.1) [33].

## Results

### Characteristics baseline details of 195 respondents

Data may be categorized depending on its Imbalance Ratio (IR), which is defined as the relation between the majority and minority class instances, by the expression $IR = \frac{N^-}{N^+}$ where $N^-$ is the number of instances belonging to the majority class and $N^+$ is the number of instances belonging to the minority class. We consider that a IR above nine represents a high IR in a data set, as ignoring the minority class instances by a classifier supposes an error of 0.1 in accuracy which has poor relevance [35].

The characteristics baseline details were shown in Table 1, among the 195 samples, the seven characteristics' IR were almost below 4 (except the degree of self-report of the disease).

### Decision tree analysis

Fig 1 presents three findings regarding outpatients' healthcare-seeking behaviors. First, for the tree, whether outpatients have medical insurance was selected as the most discriminant variable to differentiate "yes" and "no" at the first split, with those without medical insurance being more likely to visit small hospitals for minor diseases and large ones for serious diseases. In addition, among those who do have medical insurance, outpatients' profession was selected

**Table 1. Seven outpatients characteristics (N = 195).**

| Variable | n | % | Variable | n | % | Variable | n | % |
|---|---|---|---|---|---|---|---|---|
| **Ge** | | | **MI** | | | **RL** | | |
| male (0) | 85 | 43.59 | yes (0) | 155 | 79.49 | in Jiaxing (0) | 143 | 73.33 |
| female (1) | 110 | 56.41 | no (1) | 40 | 20.51 | near Jiaxing (1) | 52 | 26.67 |
| **Bi** | | | **DSr** | | | **Ag** | | |
| in Jiaxing (0) | 82 | 42.05 | minor (0) | 76 | 38.97 | <40 (0) | 54 | 27.69 |
| near Jiaxing (1) | 57 | 29.23 | medium (1) | 100 | 51.28 | 40–60 (1) | 73 | 37.44 |
| from a distance(2) | 56 | 28.72 | serious (2) | 19 | 9.75 | >60 (2) | 68 | 34.87 |
| **Pr** | | | | | | | | |
| farmer (0) | 18 | 9.23 | staff or civil servant (2) | 64 | 32.82 | | | |
| worker (1) | 57 | 29.23 | freelancer or individual owner(3) | 56 | 28.72 | | | |

The number and percentage of seven categorical variables: Gender(Ge), Medical Insurance(MI), Residency Location(RL), Birthplace(Bi), Degree of Self-report of the disease(DSr), Age(Ag) and Profession(Pr).

as the second-most discriminant variable at the second split. That is, staff or civil servants and freelancers or individual owners are more likely to visit large hospitals for minor and serious diseases, while industrial workers are more inclined to visit small hospitals for minor diseases and large hospitals for serious ones. Moreover, if outpatients were farmers, location was selected as the third-most discriminant variable for differentiating "in Jiaxing" and "near Jiaxing" at the third split. For the residing "near Jiaxing", they were more likely to choose small hospitals for minor diseases and large hospitals for serious ones.

Fig 1 (i) The 0 branch of insurance (medical insurance) denotes "yes," and 1 denotes "no"; (ii) The 0, 1, 2, and 3 branches of profession denote farmer, industrial worker, staff or civil servant, and freelancer or individual owner, respectively; (iii) The 0 branch of location (residency location) denotes in Jiaxing, while 1 denotes near Jiaxing; (iv) The 0 branch of born (birthplace) denotes Jiaxing, 1 denotes near Jiaxing, and 2 denotes distant from Jiaxing; (v) The 0 branch of age denotes children or youth, 1 denotes middle age, and 2 denotes old age; (vi) Outpatients attended a large hospital regardless of the disease's severity (option A); (vii) Whether outpatients visited a large or small hospital for serious or minor disease, respectively (option C).

## Logistic regression analysis

The last model was $logit(p_i) = 0.206 + 0.946Si—0.824P-3_i$. (P-3 was a dummy variable representing freelancer or individual owner). The likelihood ratio was $\chi^2 = 9.92$, p value = .007, therefore, the overall model was statistically significant. Regarding S and P-3 independent variables, their p values were 0.017 and 0.016, respectively. In logistic regression, another important value is the odds ratio (OR); the OR values for S and P-3 were 2.576 and 0.439, respectively. This means that among individuals with the same profession, the OR of patients without medical insurance (who were more likely to choose option C than A in the questionnaire) was 2.576 times more. Furthermore, within the same condition of having or not having medical insurance, the OR of patients who were not freelancers or individual owners (who were more likely to choose option C than A) was 1/0.439 = 2.28 times more (Table 2).

## Analysis of data from the population health platform

After removing duplicate patients and invalid samples, 850 and 830 cases from dataset 1 (the tertiary hospitals) and 2 (the primary and secondary hospitals), respectively, were separately selected. Comparing the two datasets showed that there was no significant difference in gender and age (p value >.05), however, the incidence of having medical insurance ($\chi^2 = 10.802$, p value = .001) and non-farmers' profession in dataset 1 ($\chi^2 = 52.062$, p value = .000) was higher than that in 2. We then analyzed whether the farmer profession was related to medical insurance. In dataset 1, under the condition of no gender and age difference between the farmer and non-farmer groups (p value >.05), the incidence of not having medical insurance and

**Table 2. Results of logistic regression [a].**

|  | Odds Ratio | Std. Err. | z | P>\|z\| | [95% Conf. Interval] | |
|---|---|---|---|---|---|---|
| P-3[b] | 0.438 | 0.149 | -2.42 | 0.016 | 0.225 | 0 .855 |
| S[c] | 2.576 | 1.020 | 2.39 | 0.017 | 1.185 | 5.599 |
| Constant | 1.228 | 0.221 | 1.14 | 0.253 | 0.863 | 1.748 |

[a] Number of obs = 195, LR chi2 (2) = 9.92, Prob > chi2 = 0.0070.

[b] P-3: Freelancer or individual owner; [c] S: Having or not having medical insurance.

**Table 3. Comparison of outpatients' target data between the group of tertiary hospitals and that of primary and secondary hospitals groups.**

| Target | In the tertiary hospital group ($n = 850$) | In the primary and secondary hospital group (n = 830) | The value of $\chi^2$ | P value |
|---|---|---|---|---|
| Male / female (number of cases) | 385/465 | 374/456 | $\chi^2 = 0.009$ | 0.923 |
| Children or youth (<40) / middle age (40–59) / old age (>59) (number of cases) | 423/180/247 | 449/171/210 | $\chi^2 = 3.764$ | 0.152 |
| With medical insurance / without medical insurance (number of cases) | 542/308 | 464/366 | $\chi^2 = 10.802$ | 0.001 |
| Farmer / non-farmer (number of cases) | 301/549 | 439/391 | $\chi^2 = 52.062$ | 0.000 |
| Target | Farmers in the tertiary hospital group ($n = 301$) | Non-farmers in the tertiary hospital group ($n = 549$) | | |
| Male / female (number of cases) | 123/178 | 262/287 | $\chi^2 = 3.692$ | 0.055 |
| Children or youth (<40) / middle age (40–59) / old age (>59) (number of cases) | 144/65/92 | 279/115/155 | $\chi^2 = 0.749$ | 0.688 |
| With medical insurance / without medical insurance (number of cases) | 160/141 | 390/159 | $\chi^2 = 27.221$ | 0.000 |

being a farmer was higher than that of non-farmer patients ($\chi^2 = 27.221$, p value = .000; Table 3).

We concluded that for minor or general diseases, outpatients without medical insurance and employed as farmers were more likely to choose primary and secondary than tertiary hospitals, which was mostly consistent with the aforementioned results from the decision tree model. This difference may be because of the farmer or non-farmer profession, as the proportion of farmers with medical insurance is lower than that of other professions.

## Discussion

With the goal of informing policy to effectively implement the THDS in China in the future, we explored outpatients' healthcare-seeking behavioral decision-making and its relationship with their characteristics reflecting sociodemographic and beliefs. Accordingly, we collected data about outpatients' characteristics and decision-making by questionnaires and analyzed them using a decision tree model and logistic regression. Moreover, to verify our conclusions, we randomly selected outpatient data from the regional population health platform for further statistical analysis.

We used a decision tree's machine learning method to draw "portraits" of outpatients based on their sociodemographic and belief characteristics, which provided detailed decision routes that the logistic regression could not supply. However, the logistic regression analysis produced consistent results. In this study, the decision tree and logistic regression revealed similar results as the existing research [32], which were also consistent with the data analysis from the population health platform of Jiaxing City.

First, the results from the three methods revealed that having/not having medical insurance was the primary factor that guided the outpatients' decision regarding which medical institution they visited: those with medical insurance and minor disease were more likely to choose tertiary hospitals, while those without medical insurance and minor diseases were more likely to choose primary and secondary hospitals. If the latter's choices were for economic factors, those who had medical insurance and abandoned it were more due to economic reasons. It is noteworthy that, in recent years, according to Liu P [36], a higher proportion of medical expenditure is reimbursed by insurance payments in China significantly, however, the participation rate in insurance programs has declined. This is also evident from the participation rate of the total permanent population of Jiaxing that decreased from 86.1% in 2019 to 78.4% in

2020 [37]. This shows that for residents who usually do not see a doctor or occasionally visit one for a minor illness, it is not cost-effective to pay the insurance premium compared with the proportion that can be reimbursed (See section 2 in S1 File for more details).

Moreover, the analysis revealed that, among those with medical insurance, their profession, which is representative of social class in China, was the second most influential factor guiding outpatients' healthcare-seeking behaviors. For example, staff and civil servants, as well as free-lancers and individual owners have greater advantages than farmers and industrial workers regarding economic power and individual capabilities. Therefore, they will choose tertiary hospitals with impression for better diagnosis and treatment expertise, regardless of minor or serious diseases. In contrast, farmers and industrial workers are more inclined to avoid seeking a more costly standard of care.

A similar conclusion can also be drawn from the analysis of the data from the population health platform: outpatients who were employed as farmers were more likely to choose primary and secondary hospitals. Professional differences were the same as the differences in having/not having medical insurance. In China's cities, a considerable number of migrant workers are farmers in their place of origin. In recent years, although the medical insurance coverage rate of Chinese residents has been significantly improved compared with that of more than 10 years ago [38], and migrant workers may have medical insurance in their places of origin, they do not receive medical insurance because the cross-regional reimbursement of medical insurance has not yet been realized between different provinces and cities. The analysis of the population health platform revealed a gap in the availability of medical insurance between farmers and non-farmers in tertiary hospitals: people with better paying professions were more likely to be able to afford insurance. These findings are consistent with international research showing that income is an important determinant of medical accessibility, especially in countries that lack universal medical insurance and predominately rely on private medical insurance [39] (See section 2 in S1 File for more details).

Economic factors were the most important for outpatients' healthcare-seeking behavioral decision-making in China, despite having or not having medical insurance or belonging to a specific profession. As to patients' beliefs, we can see from the questionnaire data: 19 outpatients whose self-recognition were serious diseases all chose tertiary hospitals to see a doctor on the day we surveyed, of which 13 chose the best two tertiary hospitals in Jiaxing, and none of them chose primary hospital. Hence, in China, regarding patients' beliefs, there is a significant difference in medical quality between tertiary and primary and secondary hospitals; thus, patients are willing to choose tertiary hospitals for minor and serious diseases. However, due to financial limitations, some patients who do not have medical insurance or are industrial workers or farmers will choose to be transferred to primary and secondary hospitals to treat their minor diseases. Based on these, we propose that the fundamental way to solve outpatients' overcrowding in tertiary hospitals is to reduce current discrepancies among Chinese hospitals' quality of diagnosis and treatment.

However, in recent years, it has been difficult to narrow the gap in medical quality between tertiary and primary and secondary hospitals. Since 2017, the THDS reforms have attempted to increase the coverage of the elementary medical insurance population and change the ratio of medical insurance reimbursement: a reduced reimbursement ratio in tertiary hospitals and an increased ratio in primary and secondary hospitals [12]. According to our research results, these reforms would increase number and diversity of patients in primary and secondary hospitals, who are currently mostly farmers and industrial workers, which can strengthen the construction of primary and secondary hospitals. However, the economic means would not decrease the number of patients in tertiary hospitals, who are mostly wealthy people working as freelancers or individual owners.

This indicates that the effective implementation of the THDS policy is limited by patients' freedom to choose the hospital or doctor. In the future, the government needs to formulate reform plans for healthcare providers, such as payment methods for medical insurance. At present, medical insurance payment in China is mainly based on service items, while in developed countries it is based on disease types. There has also been a growing interest in using diagnosis-related groups (DRGs) payment to reimburse inpatient care worldwide [40]. In Zhejiang Province, China, for example, DRGs payments started in January 2020, with medical insurance paid according to the administrative coordination area [41]. This can increase competitiveness among hospitals in the coordination area and guide patients with serious diseases to tertiary hospitals, diverting patients with minor diseases to primary and secondary hospitals.

Moreover, according to the decision tree model for farmers with medical insurance, diverting patients to primary or secondary hospitals is easier to achieve for outpatients living near Jiaxing. Long distances and inconvenience are factors outpatients consider when they seek medical treatments. This is consistent with the findings in the United States [42], Australia [43], and Turkey [44], which have confirmed a negative relationship between distance and readmission rate, and a positive one between distance and length of hospital stay. This also suggests that medical regional administrators could strengthen outcomes by building primary and secondary hospitals near urban cities to retain more patients and reroute patients to these hospitals.

Our findings further revealed that the outpatients' belief was an important factor influencing their health-seeking decision. Primary and secondary hospitals are competent when dealing with minor or general diseases, however, people's belief leads to prejudice, causing them to prefer visiting tertiary hospitals under better economic conditions. Therefore, government departments need to correct this bias of residents. They can guide them by disclosing the treatment performance of the second and primary hospitals, and publicizing the effective treatment of chronic and general diseases in these hospitals, rather than only the treatment results of tertiary hospitals for difficult and rare diseases.

Finally, through this applied research, we found that due to the lack of data cleaning, standardized processing, and algorithm analysis suitable for the platform, it was difficult to conduct an adequate analysis of at the population data level. For example, the platform research was conducted by random sampling, and the profession data were imperfect. Hence, access to good health system data remains one of the biggest barriers to developing effective policies in the area. Next, we should enhance the building of the population health platform.

## Supporting information

**S1 File.**
(DOCX)

**S1 Appendix.**
(DOCX)

**S1 Data.**
(XLSX)

## Acknowledgments

We would like to thank Editage (www.editage.com) for English language editing.

## Author Contributions

**Conceptualization:** Cheng Jiang.

**Data curation:** Zan Yang, Cheng Jiang.

**Formal analysis:** Lemin Shi.

**Supervision:** Lemin Shi.

**Validation:** Mingming Yu.

**Writing – original draft:** Mingming Yu.

**Writing – review & editing:** Mingming Yu.

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
