## [Decision Letter · Decision Letter 0]

23 Nov 2021

PONE-D-21-20402

The Relationship between Outpatients’ Sociodemographic and Psychological Characteristics and their Healthcare-Seeking Behavioral Decision-Making: Evidence from Jiaxing City, China

PLOS ONE

Dear Dr. Yu,

Thank you for submitting your manuscript to PLOS ONE. After careful consideration, we feel that it has merit but does not fully meet PLOS ONE’s publication criteria as it currently stands. Therefore, we invite you to submit a revised version of the manuscript that addresses the points raised during the review process. My apologies for the delay in the review process. It was difficult to obtain responses from qualified reviewers.

The authors differ substantially in their evaluations of the manuscript. However, both reviewers agree on some issues, which I think deserve special attention.

The first is more detailed information that reconciles the data in the manuscript with the national and local health system. Reviewer 1 one wants more information about an apparent discrepancy in insurance coverage in the sample with statistics on insurance coverage (or perhaps statutory insurance coverage). Reviewer 2 wants more information on how the data relate to the structure of hospital intake and delivery with respect to primary and tertiary care hospitals.

Both reviewers want more information on the methods used to estimate and validate the decision tree, and have some questions about your evaluation of the results of the validation step, and interpretation of the final tree.

Reviewers also request more information on several aspects of the analysis, and these queries should all be addressed.

We look forward to receiving your revised manuscript.

Kind regards,

James M. Lightwood

Academic Editor

PLOS ONE

Journal Requirements:

2. Please upload a new copy of Figure 1 as the detail is not clear. Please follow the link for more information: " ext-link-type="uri" xlink:type="simple">https://blogs.plos.org/plos/2019/06/looking-good-tips-for-creating-your-plos-figures-graphics/"
" ext-link-type="uri" xlink:type="simple">https://blogs.plos.org/plos/2019/06/looking-good-tips-for-creating-your-plos-figures-graphics/"

3. Please include a caption for figure 1.

Reviewers' comments:

Reviewer's Responses to Questions

**Comments to the Author**

1. Is the manuscript technically sound, and do the data support the conclusions?

Reviewer #1: Partly

Reviewer #2: Partly

2. Has the statistical analysis been performed appropriately and rigorously? 

Reviewer #1: Yes

Reviewer #2: Yes

3. Have the authors made all data underlying the findings in their manuscript fully available?

Reviewer #1: No

Reviewer #2: Yes

4. Is the manuscript presented in an intelligible fashion and written in standard English?

Reviewer #1: No

Reviewer #2: Yes

5. Review Comments to the Author

Reviewer #1: Encouragingly, the authors adopted a new analytical ideas and combined methods to analyze the behavior of medical services. However, these still can not make up for several important mistakes in this paper, so that I had to seriously consider the practical significance of the article.

First, there were misunderstanding of socio-demographic and psychological characteristics factors, only few factors like gender, age, occupation type, were included in the research, which were just the tip of an iceberg of many possible influences. And the authors did not explicitly explain why these had been chosen as representatives.

Second, the author's understanding of medical insurance seemed to be quite different from the national basic situation. In China, the basic health insurance systems included 3 parts as basic employee health insurance, basic residents health insurance, the new rural cooperative medical care system (had been consolidated into the basic resident health insurance), and the coverage is over 95% of China’s residents. But in this study, only half of them had health insurance. Why? How could it happen? And most of the findings were from the discussion of whether they have insurance.

Third is the sampling design. There were no details for the sampling, such as the calculation of sample size, sampling method, sampling time were missing. Also, the baseline details of the health database in the third part of analysis were missing. These problems leaded to a decrease in the credibility of the article.

Forth, what is artificial intelligence? Is decision tree research AI? If the author don not want to use the AI tools and is going to discuss it, don't put any emphasis on artificial intelligence.

Last, the severity of the disease was determined by who. The author did not give a more accurate criterion or judgment, I cannot judge if it is not clinicians, this judgment has any meaning and value. There's no way to accurately determine the severity of a disease just by looking at the name of the disease. Cold also can cause serious problems, that also is why the DRGs and CMI were used in these days.

Reviewer #2: This is an interesting and important study, looking at an issue that affects every hospital worldwide - that is, why do patients seek an inappropriate level of healthcare at tertiary centers for non-urgent or minor care? In Western countries, this heath seeking behavior is noted in emergency departments, and much research is being conducted to find ways of diverting these patients to lower levels of care. This behavior is complex and multilayered with many factors contributing including those that can

classified as individual-level, health system-level, and community-level factors. Therefore the authors are to be commended for trying to unpick some of the characteristics of the healthcare seeking population of Jiaxing, in order to identify the factors that drive this overuse of tertiary care.

Firstly, I would query the use of 'psychological' in the title and would recommend its removal - psychological relate to the mental and emotional state of a person, whilst this study is examining the 'belief system' of patients, which is slightly different.

Methodology

One of my greatest concerns with this study is that it appears that the city of Jiaxing is predominantly serviced by tertiary hospitals. At first read I had assumed that the study had chosen 5 tertiary and secondary hospitals and only one primary, however these six healthcare centers represent 100% of all outpatient visits in Jiaxing (page 8 and 9). Capacity would therefore appear to be the main issue - does the one primary care center have capacity to deal with all of the patients that go to tertiary centers? This goes to community level factors - what is the availability of services including health care (24 hours or limited access? primary or community services available?) but also including public transportation (do they simply go to the nearest center because it is too difficult to go to the more appropriate center?) If this is correct, then this issue should be explored more. If I have this wrong, then the text may need revising.

This leads into my next question regarding the questionnaire, which asks patients if they live in Jiaxing (0) or near Jiaxing (1). The question should be one that relates to both individual and health system level factors - what is the location of the patient's residence in relation to the hospital they are attending? What is the distance they have travelled to get to the hospital? How did they travel to the hospital? Page 20 of the manuscript show that the authors have been thinking about this question as they discuss research that has demonstrated that long distances and inconvenience are factors that outpatients consider when seeking medical treatment.

The decision tree analysis is interesting (complex!) and could potentially yield some very useful data. The first node is insurance/no insurance - however gender appears to be missing. Research in other countries has consistently identified women as more likely to seek health care than males. In the initial description of the outpatient survey of 195 patients (page 9) it is unclear if an even gender balance was achieved - the paper states that the sample was selected at random using outpatient ID. This may have been omitted in the decision tree analysis as when the training data set was applied to the larger outpatient data sets, there was no significant difference in gender (although slightly more females).

Conclusions

It is a shame that the large data set could not distinguish any greater detail regarding profession (farmer/non farmer) (page 12). The lack of data should have been discussed - access to good health system data is one of the biggest barriers in developing effective policy in this area. This would be worth including in the conclusion - as a where to next.

On page 18, I would recommend rewording this sentence as it is confusing 'This discrepancy between the ratios may indicate that more patients without medical insurance than with medical insurance would choose to not visit hospitals.' You cannot make a conclusion about patients who do not go to hospital - you have no data to support this conclusion. You only have data about those who do go - more patients with insurance go to hospital

At the bottom of page 18, it is stated that the profession of outpatients was the second most influential factor. This conclusion can only be drawn from the smaller study/questionnaire as the granularity of profession wasn't available in the larger study - only farmer versus non farmer. This should be reworded to reflect this. As a discussion point this makes sense as better paying professions would be more likely to be able to afford insurance.

One of the most interesting statements on page 21, was the final point that patients believed that they would get better care in the larger tertiary centers (hence the title beliefs rather than psychological) - this is the real problem - how to address this belief, how to change patient behavior?

Whilst I agree with the final conclusion 'Accordingly, we propose that the government should focus on economic reforms to increase outpatients’ visits to small hospitals as well as diagnosis-related groups (DRGs) payment of medical insurance to decrease the admitting of patients with minor diseases in large hospitals.', I would have thought one of the most obvious areas to address is the availability/access to primary health care centers.

6. PLOS authors have the option to publish the peer review history of their article (what does this mean?). If published, this will include your full peer review and any attached files.

Reviewer #1: No

Reviewer #2: No

---

## [Author Response · Author response to Decision Letter 0]

3 Jan 2022

1. First, there were misunderstanding of socio-demographic and psychological characteristics factors, only few factors like gender, age, occupation type, were included in the research, which were just the tip of an iceberg of many possible influences. And the authors did not explicitly explain why these had been chosen as representatives.

Response:

Thank you for your comments that we have considered and incorporated in the revised draft of this manuscript. Accordingly, we have changed psychological characteristics to belief characteristics. You are correct in pointing out that the original manuscript did not specify how these variables were selected. In fact, during the design and research process, we undertook a detailed literature review, but could not include it in our article due to limits on the length of the article. In the revised manuscript, we have added some of the following information regarding the main selection basis on page 6,7. 

In 1968, Andersen first proposed the behavioral model of medical service utilization, which represented a systematic research into patients’ healthcare-seeking behavioral decision-making. The structural framework of the model comprises the following elements. 1) Environmental factors: They primarily include the external environment, such as natural, political, economic, and medical system. 2) Population features: They predict and explain the use of medical services through three variable levels, affecting health outcomes and service satisfaction. Predisposing factors: They can be categorized into personality factors, social structure, and health beliefs. Personality factors, such as age, gender, and other demographic factors, indicate people’s need for medical services based on their physiological characteristics. Social structure is measured by a series of indicators that reflect the individuals’ social status and their ability to deal with problems. Common indicators include education, profession, race, social network, social relations, and culture. Health beliefs refer to people’s attitudes, values, and knowledge of health and medical services. They affect people’s perception of the use and demand intensity of medical services. Enabling factors: They promote or hinder people’s ability to use medical services, including personal, family, and community resources. Personal resources include income, medical insurance, and time to arrive at medical institutions among others. Family resources include family structure and function. Community resources are mainly supplier factors. Need: It is the direct cause of medical service utilization. Affected by social factors and health beliefs, it reflects how people view their health and functional status, experience disease symptoms and pain, and worry about their health. When necessary, self-assessment (subjective needs) and assessment (objective needs) can be carried out. 3) Health behavior: It includes personal medical and health service utilization behavior. 4) Health outcome: It can be evaluated using three aspects: personal self-perception of health status, health status assessed by professional medical personnel, and patient satisfaction [1]. 

Using the above framework, many scholars have examined the influencing factors of individual aspects, such as Andersen pointed out that symptoms themselves are also a form of social construction. Patients’ perceptions and interpretations of symptoms are affected by their social and cultural backgrounds [2]. In addition, some researchers believe that a relatively “disadvantaged group” is more inclined to self-diagnosis, choosing a hospital closer to their residence, or a private hospital [3]. Some scholars in China have indicated that patients tend to choose a high-level medical institution given sudden or severe illness; greater financial resources enable patients to actively seek health care services, and the improvement in cultural literacy facilitates patients paying more attention to their own health [4].

 Moreover, Behavioral Decision Theory posits that the emergence of any decision is inseparable from three factors: the contextual factors underlying the decision, the features of the individual’s beliefs, and the individual’s preference structure [5,6]. According to the theoretical background of Behavioral Decision Theory, the American psychologist Daniel Kahneman and the Israeli behavioral finance scientist Amos Tversky [7] proposed a new theory of behavioral decisions, Prospect Theory, by introducing psychology to economics. Prospect Theory is based on the individual’s actual state of decision-making; it focuses on the psychological reasons for their behavior. According to Prospect Theory, subjective systematic bias and individual risk preference directly affect the decision-making process and the patient’s interpretation and value judgment of the expected goal, thus affecting their healthcare-seeking behavioral decision-making. Subjective systematic bias varies with cognitive ability and disease severity. For example, when patients face uncertain prospects at decision-making nodes, the more serious they consider their illness, the greater their fear, and the greater the extent their decision-making will be affected by their risk preference, resulting in less informed or objectively based healthcare-seeking behaviors [7]. In contrast, if the expected result is highly certain, patients will comprehensively consider factors including medical technology, economy, and convenience of medical treatment, and their healthcare-seeking behaviors will tend to be more reasonable [8].

According to the theoretical framework described above and the data available from the questionnaires, we examined six patient sociodemographic characteristics：gender, age, profession, birthplace, location of residence, and having medical insurance. Additionally, using the Prospect Theory paradigm and health beliefs or outcomes, we evaluated patients’ self-recognition of the severity of their disease (serious, medium, or minor). It is important to clarify that patients’ financial status is very important in our research. However, to respect patients’ privacy, their profession was used as a proxy (e.g., farmer, industrial worker, staff or civil servant, freelancer or individual owner) given that, in China, profession and income or social stratum are correlated to some extent [9].

Reference:

[1](America) Wallinsky.(1999). Sociology of Health (Sun, M.H., Trans). Beijing China, BJ: Social Sciences Literature Press.

[2]Andersen, R.S., Vedsted, P., Olesen, F. (2009). Patient delay in cancer studies: a discussion of methods and measures. BWC Health Service Research,9(189).

[3] Puck, D.C., Beukers, Ron, G.M., Kemp, M. V. (2014). Patient hospital choice for hip replacement: empirical evidence from the Netherlands. Eur J Health Econ, 15, 927-936.

[4] Guo, W.Q., Wu, Y.A., Yao, Z.Y. (2010).Analysis of medical behaviors and influencing factors of rural chronic disease patients. Chinese Primary Health Care, 24 (1), 65-67.

[5] George, L., Jennifer, L. (2003).The Role of Affect in Decision Making: Handbook of Affective Science (pp.256-298). Oxford England, Oxford: Oxford University Press. 

[6] Herbert, G. (2007).A framework for the unification of the behavioral sciences. Behavioral and Brain Sciences, 30(1), 1-61.

[7] Daniel, K., Amos, T. (1979). Prospect Theory: an analysis of decision under risk. Econometrica, 47(2), 263-292.

[8] Huang, H. (2010). Study on health seeking behavior of community residents (pp.14-15).[Unpublished Master's thesis]. Jiangsu University, China.

[9] Xueyi Lu. (2018). Research Report on social structure of contemporary China. Beijing China, BJ: social sciences academic press.

2.Second, the author's understanding of medical insurance seemed to be quite different from the national basic situation. In China, the basic health insurance systems included 3 parts as basic employee health insurance, basic residents health insurance, the new rural cooperative medical care system (had been consolidated into the basic resident health insurance), and the coverage is over 95% of China’s residents. But in this study, only half of them had health insurance. Why? How could it happen? And most of the findings were from the discussion of whether they have insurance.

Response:

Thank you for your comments. Zhu Chen, Minister of Health of China, announced at the second China Health Forum in 2011 that the medical insurance coverage rate of Chinese residents had increased from about 15% in 2000 to nearly 95% at the end of 2010, covering 1.27 billion people [1]. However, this is the whole country’s data, which cannot be used to assume that all regions are the same. In addition, in 2010 and after more than 10 years, most parts of the country have still not ensured that an outpatient’s settlement of medical insurance in a different city be affiliated to their origin city. In other words, for example, although the permanent nonnative outpatients who saw a doctor in Jiaxing had medical insurance in their native city, they could not be reimbursed by Jiaxing’s system of medical insurance. The following Table 1 presents changes in the number and insured rate of permanent residents in Jiaxing, including the number of Jiaxing’s native and nonnative residents who have participated in medical insurance in recent years.

Table 1. The number and insured rate of permanent residents in Jiaxing 

 Permanent native population in Jiaxing Permanent nonnative population in Jiaxing Total permanent population in Jiaxing Total Jiaxing-

insured population

in Jiaxing Jiaxing-

insured population of Jiaxing’s native Total 

Jiaxing-

insured rate Jiaxing-

insured rate of

Jiaxing’s 

native Jiaxing-

insured rate of

Jiaxing’s 

nonnative 

2016 352.1 109.3 461.4 248.7 208.5 53.9% 59.2% 36.8%

2017 356.4 109.2 465.6 261.8 216.5 56.2% 60.8% 41.5%

2018 360.4 112.2 472.6 403.1 354.7 85.3% 98.4% 43.2%

2019 363.7 116.3 480.0 413.3 362.1 86.1% 99.6% 44.1%

2020 367.6 172.5 540.1 423.5 366.6 78.4% 99.7% 33.0%

Data sources: the statistical yearbook on the official website of Jiaxing Statistics Bureau, the annual medical insurance express on the official website of Jiaxing Medical Insurance Bureau, and internal data of the government.

Table 1 shows that for the total permanent population in Jiaxing (including native and nonnative), the rate of participating in Jiaxing local medical insurance has increased yearly from 53.9% in 2016 to 78.4% in 2020; however, the rate of nonnative residents’ participating in Jiaxing local medical insurance was low every year, no more than 50%, with some of them participating in the medical insurance in their cities of origin. According to the staff of Jiaxing Medical Insurance Bureau, Jiaxing did not implement the reimbursement of outpatient’s medical insurance to be affiliated to different provinces and cities until December 2020, implying that although the nonnative population in Jiaxing had medical insurance in their city of origin, it was generally impossible for outpatients in Jiaxing to realize its use before 2020. This study’s questionnaire survey was conducted in July 2018, and the outpatient data extracted from Jiaxing population health platform mainly came from 2017–2019. In 2017, the insured rate of the total permanent population of Jiaxing was only 56.2%, and that of Jiaxing’s native residents was only 60.8%. In addition, in the field survey, it was found that even if several outpatients have medical insurance, in view of the high starting line of medical insurance reimbursement, or the high percentage of the billed charges being paid by the patients themselves, some outpatients, especially those who do not spend too much money to see a doctor occasionally, simply pay at their own expense, or stop participating in the insurance the next year. According to the “statistical express on the development of medical security in 2020” recently released by the National Medical Insurance Bureau, the number of urban and rural residents participating in basic medical insurance in 2020 decreased by 8.06 million, or 0.8% lower compared with 2019 [2], similarly, the participation rate of the total permanent population of Jiaxing also decreased from 86.1% in 2019 to 78.4% in 2020. In addition, according to Ping Liu [3], while the proportion of medical insurance reimbursement in China has increased significantly, the participation rate is declining. According to the statistics of the relevant departments of Zhejiang Province, at the end of 2016, the cooperative medical fund for urban and rural residents in Nanhu District of Jiaxing City reported a cumulative loss of 7.59 million yuan. This could seriously affect the development of urban and rural medical reform, resulting in the phenomenon of “cooking out” of medical security in the next few years.

We have added this information in the discussion section of the revised manuscript on page 20,21.

Reference:

[1]Zhu Chen.(2011). China's basic medical insurance coverage rate is nearly 95%, covering 1.27 billion people. Decision making reference of hospital leaders, 17(03).

[2]National Medical Insurance Bureau.(2021). Statistical bulletin on the development of medical security in 2020.

[3]Ping Liu. (2018). Problems and Countermeasures of medical insurance for urban and rural residents. Rural economy and technology, 29 (08): 216.

3.Third is the sampling design. There were no details for the sampling, such as the calculation of sample size, sampling method, sampling time were missing. Also, the baseline details of the health database in the third part of analysis were missing. These problems leaded to a decrease in the credibility of the article.

Response:

Thank you for your valuable comments. We have added the missing parts to the revised manuscript on page 9. 

The size of the questionnaire sample was 200 patients, calculated using the following formula:, where, Z score is 1.96 when the degree of confidence is 95%, and P is the probability value, generally 0.5, and E is the sampling error rate taken as 7%.

The sampling method is discussed on page 10 and 11 of the revised manuscript. 

To ensure that the sample was randomly selected, in given hospitals, all questionnaires were administered simultaneously (July 5, 2018) and the respondents were selected using a random sampling approach based on outpatient ID number, without excluding any disease. The outpatients were instructed to complete the questionnaire based on their general health-seeking behaviors. Only three and two outpatients in the tertiary and primary hospitals, respectively, refused to participate. Thus, 195 valid questionnaires were obtained. The survey process was approved and supervised by the Health Commission of Jiaxing, and all participants signed written informed consent forms before completing the questionnaire.The baseline details of the 195 respondents are presented in the following Table 2.

Table 2. Seven outpatients characteristics (N=195)

Variable n % Variable n % Variable n %

Gender Medical insurance Residency location 

 male (0) 85 43.59 yes (0) 155 79.49 in Jiaxing (0) 143 73.33

 female (1) 110 56.41 no (1) 40 20.51 near Jiaxing (1) 52 26.67

Birthplace Degree of self-recognition of the disease Age 

 in Jiaxing (0) 82 42.05 minor (0) 76 38.97 40 (0) 54 27.69

 near Jiaxing (1) 57 29.23 medium (1) 100 51.28 40–60 (1) 73 37.44

from a distance (2) 56 28.72 serious (2) 19 9.75 60 (2) 68 34.87

Profession 

 famer (0) 18 9.23 

 worker (1) 57 29.23 

 staff or civil

servant (2) 64 32.82 

 freelancer or individual owner (3) 56 28.72 

4.Forth, what is artificial intelligence? Is decision tree research AI? If the author don not want to use the AI tools and is going to discuss it, don't put any emphasis on artificial intelligence.

Response:

Artificial intelligence is a branch of computer science. It attempts to understand the essence of intelligence and produce a new intelligent machine that can respond in a manner similar to human intelligence. Research in this field includes robots, language recognition, image recognition, natural language processing, and expert system. Machine learning is only a subset of artificial intelligence. It obtains the information needed by human beings through algorithm learning from data. Machine learning algorithms can be categorized as supervised learning, unsupervised learning, semi supervised learning, and reinforcement learning. Decision tree is a non-parametric, supervised machine learning method. Machine learning is concerned with features and labels. In the present research, the main machine learning task was to extract useful features and construct the mapping from features to labels; in other words, to draw a portrait.

Therefore, thank you for pointing out that the original manuscript blurs the difference between machine learning and artificial intelligence. To rectify this error, AI has been changed to machine learning in the revised manuscript on page 6,7.

5.Last, the severity of the disease was determined by who. The author did not give a more accurate criterion or judgment, I cannot judge if it is not clinicians, this judgment has any meaning and value. There's no way to accurately determine the severity of a disease just by looking at the name of the disease. Cold also can cause serious problems, that also is why the DRGs and CMI were used in these days.

Response:

First, comprehensive clinical diagnosis includes not only the diagnosis of etiology, pathoanatomy, and pathophysiology, but also the classification and staging of diseases, as well as the diagnosis of complications and associated diseases. The classification or staging of “severe” or “middle and late stage” was not reflected in the diagnostic names of “minor diseases” or “common diseases” selected from Jiaxing population health platform, and there were no complications and associated diseases.

In addition, we would like to thank the reviewer for pointing out that the terms “serious disease,” “minor disease,” and “common disease” used in the manuscript are subjective concepts, lacking clinical diagnosis basis. Indeed, in the study of Jiaxing population health platform, the author subjectively defined “minor diseases” and “common diseases” based on her experience as an attending physician, and the definitions were also corroborated by the policy documents issued by the State or the Medical Insurance Bureau. This judgment was based on a statement by Shanchang Xu, deputy director of the Medical Reform Office of the State Council, who pointed out that the “serious disease” mentioned in the “Guidance on carrying out serious illness insurance for urban and rural residents” [1] was not a medical disease concept. The document simply did not distinguish serious diseases according to the types of diseases, instead, it determined them according to the comparison between the high medical expenses caused by serious diseases and the economic affordability of urban and rural residents. As long as the medical expenses paid by the insured exceed the local annual per capita income, they can enjoy serious illness insurance [2]. At present, there is no clear and unified definition of “serious illness” of the “serious illness medical insurance” program in China. According to the disease definition of major disease insurance jointly formulated by China Insurance Industry Association and China Medical Association [3], 25 serious diseases have been identified (the specific figures are still being updated), including malignant tumors (excluding some early malignant tumors), acute myocardial infarction, sequelae of stroke (permanent dysfunction), and major organ transplantation or hematopoietic stem cell transplantation (allogeneic transplantation) among others.

Thus, the revised manuscript (on page 15) emphasizes that the definition of “minor diseases” or “common diseases” has been excluded according to the scope specified by the State in the serious disease insurance program. Further, it emphasizes that this study focused on the medical decision-making behavior of outpatients, and their social consciousness level was the basis for determining “minor disease” or “common disease,” rather than scientific clinical diagnosis.

Reference:

[1]Xinhua News Agency.(2012, Aug 30th). The guiding opinions on carrying out serious illness insurance for urban and rural residents. Retrieved Aug 30th, 2012, from http://www.gov.cn/jrzg/2012-08/30/content_2213783.htm

[2]China.com.(2012, Sep 4th). Medical reform office interprets "serious illness medical insurance", which does not distinguish between diseases.Retrieved Sep 4th, 2012, from http://roll.sohu.com/20120904/n352321089.shtml

[3] China Insurance Industry Association and China Medical Association.(2020). Specification for use of disease definition of major disease insurance (revised in 2020).

Reviewer #2: This is an interesting and important study, looking at an issue that affects every hospital worldwide - that is, why do patients seek an inappropriate level of healthcare at tertiary centers for non-urgent or minor care? In Western countries, this heath seeking behavior is noted in emergency departments, and much research is being conducted to find ways of diverting these patients to lower levels of care. This behavior is complex and multilayered with many factors contributing including those that can classified as individual-level, health system-level, and community-level factors. Therefore the authors are to be commended for trying to unpick some of the characteristics of the healthcare seeking population of Jiaxing, in order to identify the factors that drive this overuse of tertiary care.

1.Firstly, I would query the use of 'psychological' in the title and would recommend its removal - psychological relate to the mental and emotional state of a person, whilst this study is examining the 'belief system' of patients, which is slightly different.

Response:

Thank you for your valuable comments. As per your recommendation, we have changed psychological characteristics to belief characteristics. In addition, we added the questionnaire survey data to support that the belief characteristics of outpatients affect the health-seeking decision-making: 19 of 195 outpatients whose self-recognition were serious diseases all chose large hospitals to see a doctor at the day we surveyed, of which 13 chose the best two tertiary hospitals in Jiaxing, and none chose community hospital, please see it on page 22 of the revised manuscript.

Methodology

2.One of my greatest concerns with this study is that it appears that the city of Jiaxing is predominantly serviced by tertiary hospitals. At first read I had assumed that the study had chosen 5 tertiary and secondary hospitals and only one primary, however these six healthcare centers represent 100% of all outpatient visits in Jiaxing (page 8 and 9). Capacity would therefore appear to be the main issue - does the one primary care center have capacity to deal with all of the patients that go to tertiary centers? This goes to community level factors - what is the availability of services including health care (24 hours or limited access? primary or community services available?) but also including public transportation (do they simply go to the nearest center because it is too difficult to go to the more appropriate center?) If this is correct, then this issue should be explored more. If I have this wrong, then the text may need revising.

Response:

Apologies for the confusion, this part of the manuscript may not be clear. Among the many medical units in Jiaxing, we have selected six, including five tertiary and secondary hospitals. These five hospitals constitute the main force of outpatient services in the city, accounting for 79.7% of the total number of outpatient services in the city in 2017; the annual outpatient visits of all primary hospitals and other secondary hospitals accounted for 15.2% and 5.1%, respectively. Because other secondary hospitals account for very little (5.1%), we did not choose the corresponding institution, however, we chose Nanhu Community Health Center, which is one of the representative primary hospitals, not the one accounting for 15.2% of all outpatient visits in 2017. At present, the three-level referral system has not been strictly implemented in the outpatient clinics in China, that is, people can choose any level of hospital (level one, two, or three) for their first outpatients’ service.

When selecting medical institutions for the questionnaire survey, our principle was that the proportion of outpatients’ visiting the hospital throughout the year should be as large as possible, and the proportion of the selected hospitals should try to conform to the distribution of real outpatients in Jiaxing throughout the year. If a hospital was selected at random, the sample data would not have represented the real overall data. In the revised manuscript on page 10, we further explain the research background in detail.

3.This leads into my next question regarding the questionnaire, which asks patients if they live in Jiaxing (0) or near Jiaxing (1). The question should be one that relates to both individual and health system level factors - what is the location of the patient's residence in relation to the hospital they are attending? What is the distance they have travelled to get to the hospital? How did they travel to the hospital? Page 20 of the manuscript show that the authors have been thinking about this question as they discuss research that has demonstrated that long distances and inconvenience are factors that outpatients consider when seeking medical treatment.

Response:

Thank you for your expert comments and for pointing out that the original manuscript did not provide a detailed introduction when discussing the outpatient’s residence. We have included supplementary explanations in the revised manuscript on page 8.

The urban area in European and American countries is called urban, a place where ordinary people go to work. The place where they go back to sleep and live after work is called suburban, and together the two are called metro; China’s cities not only have central urban areas, but also include relatively remote counties and county-level cities. For example, Jiaxing governs two districts, two counties, and three county-level cities. Some people not only work, but also live in the central urban area, that is, the urban districts, and were categorized as living “in Jiaxing (0)” in our questionnaire survey, while those “near Jiaxing (1)” generally work and live in the surrounding counties and county-level cities, and the hospitals we chose were in the central urban area of Jiaxing. Currently, the means of transportation at their disposal include walking, taking buses, mopeds, and cars. Outpatients living in the surrounding counties and county-level cities generally use buses and cars to reach the hospitals in the central urban area (of course, they can also choose the hospitals in the counties and cities close to them), with longer transportation time (usually 30 minutes to 1 hour by car, and longer by bus). Therefore, compared with outpatients living in the central urban area (it usually takes them less than 30 minutes by car), those living in surrounding counties and cities are inconveniently far away.

4.The decision tree analysis is interesting (complex!) and could potentially yield some very useful data. The first node is insurance/no insurance - however gender appears to be missing. Research in other countries has consistently identified women as more likely to seek health care than males. In the initial description of the outpatient survey of 195 patients (page 9) it is unclear if an even gender balance was achieved - the paper states that the sample was selected at random using outpatient ID. This may have been omitted in the decision tree analysis as when the training data set was applied to the larger outpatient data sets, there was no significant difference in gender (although slightly more females).

Response:

In the random questionnaire sample and the sample randomly selected by Jiaxing population health platform, it was indeed found that female outpatients are slightly more in number than men. However, according to the C4.5 algorithm of the decision tree, the information gain rates for gender and self-recognition of the severity of the disease were c. 0.1% and smaller; the contributions were very small due to cross-entropy loss. Actually, “the degree of self-recognition of the disease (serious, medium, or minor)” was the patients’ perception of that medical treatment in the outpatient department we surveyed, and the decision for choosing an option reflected the patients’ general behaviors when seeing a doctor: specifically, whether they attended a large hospital regardless of the severity of the disease (option A), or whether they visited a large or small hospital for serious or minor diseases, respectively (option C). There is no logical connection between the two. The final decision tree analysis showed that the two have nothing to do with each other, confirming the correctness of our research. In addition, gender is also a very important characteristic for some decision-makings of behavioral, but at least in our research on a certain decision-making, gender was not significant. Therefore, considering that the model has other features and that more samples are needed, we conducted another decision tree model without these two features and reanalyzed it also using the C4.5 algorithm.

Data may be categorized depending on its imbalance ratio (IR), which is defined as the relation between the majority class and minority class instances, using the following formula:, where and is the number of instances belonging to the majority and minority class, respectively. An IR above 9 represents a high IR in a data set, due to the fact that ignoring the minority class instances by a classifier supposes an error of 0.1 in accuracy, which has poor relevance [1]. Also, MLP and C4.5 (in our research, we used C4.5) are less affected by the imbalance, while SVM generally performs poorly in imbalanced problems [2]. We have included supplementary explanations in the revised manuscript on page 12,13.

Sample descriptive statistics are reported in Table 2. Among the 195 valid samples, the seven characteristics’ IR in the sample data are not all above 9. 

Reference:

[1] Lemnaru C., Potolea R. (2012). Imbalanced Classification Problems: Systematic Study, Issues and Best Practices. In: Zhang R., Zhang J., Zhang Z., Filipe J., Cordeiro J. (eds) Enterprise Information Systems. ICEIS 2011. Lecture Notes in Business Information Processing, vol 102. Springer, Berlin, Heidelberg. 

[2] García S, Herrera F. Evolutionary undersampling for classification with imbalanced datasets: Proposals and taxonomy[J]. Evolutionary computation, 2009, 17(3): 275-306.

Conclusions

It is a shame that the large data set could not distinguish any greater detail regarding profession (farmer/non farmer) (page 12). The lack of data should have been discussed - access to good health system data is one of the biggest barriers in developing effective policy in this area. This would be worth including in the conclusion - as a where to next.

Response:

We have added these details in the discussion of the revised manuscript on page 24.

On page 18, I would recommend rewording this sentence as it is confusing 'This discrepancy between the ratios may indicate that more patients without medical insurance than with medical insurance would choose to not visit hospitals.' You cannot make a conclusion about patients who do not go to hospital - you have no data to support this conclusion. You only have data about those who do go - more patients with insurance go to hospital

Response:

We have deleted this sentence from the revised manuscript.

At the bottom of page 18, it is stated that the profession of outpatients was the second most influential factor. This conclusion can only be drawn from the smaller study/questionnaire as the granularity of profession wasn't available in the larger study - only farmer versus non farmer. This should be reworded to reflect this. As a discussion point this makes sense as better paying professions would be more likely to be able to afford insurance.

Response:

We have reworded this sentence in the discussion section of the revised manuscript on page 20,21.

One of the most interesting statements on page 21, was the final point that patients believed that they would get better care in the larger tertiary centers (hence the title beliefs rather than psychological) - this is the real problem - how to address this belief, how to change patient behavior?

Response:

We have changed title “The relationship between outpatients’ sociodemographic and belief characteristics and their healthcare-seeking behavioral decision-making: evidence from Jiaxing city, China” in the revised manuscript.

Whilst I agree with the final conclusion 'Accordingly, we propose that the government should focus on economic reforms to increase outpatients’ visits to small hospitals as well as diagnosis-related groups (DRGs) payment of medical insurance to decrease the admitting of patients with minor diseases in large hospitals.', I would have thought one of the most obvious areas to address is the availability/access to primary health care centers.

Response:

We have added other areas that need addressing such as an increase in publicity for primary hospitals in the discussion section of the revised manuscript on page 24.

---

## [Decision Letter · Decision Letter 1]

11 Mar 2022

PONE-D-21-20402R1The relationship  between outpatient s ’ sociodemographic and belief characteristics and their healthcare-seeking behavioral decision-making: evidence from Jiaxing city, ChinaPLOS ONE

Dear Dr. Yu,

Thank you for submitting your revised manuscript to PLOS ONE. After careful consideration, Reviewer 2 and I feel that you have adequately addressed the reviewers' comments but the first revision does not fully meet PLOS ONE’s publication criteria as it currently stands. Therefore, we invite you to submit a second revised version of the manuscript that addresses some remaining issues. Please address the items below: 1) the additional information describing the machine learning techniques is satisfactory, however, I think the first revision includes too much detail in the text of the article, So, you went from too little in the original submission to too much in the revision in response to Reviewer 1. I suggest be edited so it can be a stand-alone amplification of the material in the main text and be incorporated with your response to Reviewer 1's comment 4 (4.Forth, what is artificial intelligence?...) and include that in a supplemental file. I've highlighted the details that I suggest could go into a supplemental file in the attached version of your revised manuscript. 2) I think Reviewer 1's comments 1 (First, there were misunderstanding of socio-demographic... ), 2 (Second, the author's understanding of medical insurance...)  and 5 (Last, the severity of the disease was determined by who...), and Revewer 2's comment 2 (One of my greatest concerns with this study...) raise important issues that should be available to readers who are interested, but are too detailed for the main text. Please edit your responses to those comments so they can be stand-along amplifications of the main text and include them, with the references, in the supplemental file that I suggested above. Please add references to this supplemental information at appropriate places in the main text. 3) Figure 1 that the software package produces won't be acceptable for publication. Please prepare another version of Figure 1 following PLOS ONE guidelines for figures. I think an acceptable figure could be prepared in Microsoft powerpoint and then converted to tif format using the PACE graphics processor (see PLOS ONE figure guidelines). 4) The data supplement requires a variable key that explains the variable names, the data values with references to the corresponding questions in the Questionnaire. In other words, it requires a brief codebook . I think this could be added as an additional tab in the spreadsheet.  5) Parts of the revised manuscript don't conform to standard English usage, so I suggest the manuscript be reviewed by someone who is fluent in academic English prose.  6) I have a number of other questions and suggestions regarding the revised manuscript.  I have attached a copy of the revised manuscript with my comments, questions and suggestions. Please address the following issues: a) The explanation for overcrowding in tertiary hospitals and policy responses in the Introduction (pp 4-5) seems to say that the 2005 THDS was adopted to address a problem created by the  2009 Health Policy Reform. Please clarify.

b) In the Theoretical Model section (pp 6-7) the theoretical model that underlies the statistical analysis seems to be conflated with the statistical model and estimation technique. Provide more justification and explanation why what seem to be three different things can be described as one 'hybrid model', or distinguish them as different components of the analysis. c) I think the relationship between the Patient Characteristics and Questionnaire Design sections is confusing and needs some reorganization or additional language to clarify d) Table 1 and its discussion should be moved the the Results section. e) Reference 27 at the bottom of page 12 (Taeho Jo, Machine Learning Foundations) describing the C4.5 algorithm isn't useful for readers not expert in machine learning (I can't find the term 'C4.5' in the text. The text does clearly use C4.5 or a closely related algorithm, but the general reader won't know this). I have suggested an alternative reference for C4.5, (and also the SVM and MLP algorithms and their relationships to decision trees), or clarify that the reference describes how that type of algorithm works, but don't suggest that the reference mentions C4.5 by name) d) I think the description on pp 13-14 of  the logistic regression analysis says that it is used to validate the decision tree estimates, but also to compare and contrast (as in a sensitivity or robustness analysis?) the results of the two methods. I think this  is confusing and frankly can't be right. I think you are using logistic regression for two distinct functions that it cannot perform simultaneously. I would like clarification of how the logistic regression analysis is used to strengthen the decision tree analysis. I think its use should be characterized as a sensitivity or robustness analysis. Please see my comments in the attachment for more details. e) The Analysis of the data from the Population Health Platform on pp 14-15 needs clarification. A clearer explanation of how it relates to the decision tree analysis is needed at the beginning of the section. I think how the Health Platform data is used to confirm the decision tree and logistic regression analysis becomes clear by the end of the presentation of the results later in the paper, but this should be made clear in the Materials and Methods section. Additional comments and suggestions are in the attached edited version of the revised manuscript.

If applicable, we recommend that you deposit your laboratory protocols in protocols.io to enhance the reproducibility of your results. Protocols.io assigns your protocol its own identifier (DOI) so that it can be cited independently in the future. For instructions see: https://journals.plos.org/plosone/s/submission-guidelines#loc-laboratory-protocols. Additionally, PLOS ONE offers an option for publishing peer-reviewed Lab Protocol articles, which describe protocols hosted on protocols.io. Read more information on sharing protocols at https://plos.org/protocols?utm_medium=editorial-emailutm_source=authorlettersutm_campaign=protocols.

We look forward to receiving your revised manuscript.

Kind regards,

James M. Lightwood

Academic Editor

PLOS ONE

Reviewers' comments:

Reviewer's Responses to Questions

**Comments to the Author**

1. If the authors have adequately addressed your comments raised in a previous round of review and you feel that this manuscript is now acceptable for publication, you may indicate that here to bypass the “Comments to the Author” section, enter your conflict of interest statement in the “Confidential to Editor” section, and submit your "Accept" recommendation.

Reviewer #2: All comments have been addressed

2. Is the manuscript technically sound, and do the data support the conclusions?

Reviewer #2: Yes

3. Has the statistical analysis been performed appropriately and rigorously? 

Reviewer #2: Yes

4. Have the authors made all data underlying the findings in their manuscript fully available?

Reviewer #2: Yes

5. Is the manuscript presented in an intelligible fashion and written in standard English?

Reviewer #2: Yes

6. Review Comments to the Author

Reviewer #2: The authors have addressed all the points raised by both reviewers adequately, and editing has improved the flow of the article.

7. PLOS authors have the option to publish the peer review history of their article (what does this mean?). If published, this will include your full peer review and any attached files.

Reviewer #2: No

---

## [Author Response · Author response to Decision Letter 1]

21 Apr 2022

1) the additional information describing the machine learning techniques is satisfactory, however, I think the first revision includes too much detail in the text of the article, So, you went from too little in the original submission to too much in the revision in response to Reviewer 1. I suggest be edited so it can be a stand-alone amplification of the material in the main text and be incorporated with your response to Reviewer 1's comment 4 (4.Forth, what is artificial intelligence?...) and include that in a supplemental file. I've highlighted the details that I suggest could go into a supplemental file in the attached version of your revised manuscript.

Response:

Thanks for your suggestions. In this revised main text, we presented the machine learning part separately in the introduction of the decision tree analysis and included a more detailed explanation in this supplemental file.

In the main text on page 11:

Machine learning is concerned with features and labels. Features are observable attributes of the observed object, and labels are what we want to predict. In this research, the main decision tree task was to extract useful features and construct the mapping from features to labels; in other words, to draw a portrait.

2)I think Reviewer 1's comments 1 (First, there were misunderstanding of socio-demographic... ), 2 (Second, the author's understanding of medical insurance...)  and 5 (Last, the severity of the disease was determined by who...), and Revewer 2's comment 2 (One of my greatest concerns with this study...) raise important issues that should be available to readers who are interested, but are too detailed for the main text. Please edit your responses to those comments so they can be stand-along amplifications of the main text and include them, with the references, in the supplemental file that I suggested above. Please add references to this supplemental information at appropriate places in the main text.

Response:

For the Reviewer 1's comments 1, 2, 5 and Reviewer 2’s comment 2, in the revised main text, we have added some information, and more details are put in the points 1 to 4 of supplemental file. Moreover, we added references to the supplemental file by endnote in the main text. 

3) Figure 1 that the software package produces won't be acceptable for publication. Please prepare another version of Figure 1 following PLOS ONE guidelines for figures. I think an acceptable figure could be prepared in Microsoft powerpoint and then converted to tif format using the PACE graphics processor (see PLOS ONE figure guidelines).

Response:

According to the requirements of PLOS ONE, we have modified the format of Figure 1. 

3)The data supplement requires a variable key that explains the variable names, the data values with references to the corresponding questions in the Questionnaire. In other words, it requires a brief codebook. I think this could be added as an additional tab in the spreadsheet. 

Response:

According to the requirements, we have added additional tabs in the spreadsheet of Table 1.

4)Parts of the revised manuscript don't conform to standard English usage, so I suggest the manuscript be reviewed by someone who is fluent in academic English prose. 

Response:

According to the requirements, we invited Editage (www.editage.com) for the English language editing.

6) I have a number of other questions and suggestions regarding the revised manuscript.  I have attached a copy of the revised manuscript with my comments, questions and suggestions. Please address the following issues:

a)The explanation for overcrowding in tertiary hospitals and policy responses in the Introduction (pp 4-5) seems to say that the 2015 THDS was adopted to address a problem created by the  2009 Health Policy Reform. Please clarify.

Response:

Thank you for your comments that we have considered and incorporated the changes in the revised main text. Before 2009 in China, the 15% mark-up policy permitted by the government on drug prices allowed public hospitals to retain 15% profits of drug prices after selling to patients. Hence, as public hospitals, they could be self-sufficient at a large cost, and financial subsidies account for a small part of their income. The 2009 Health Policy Reform included a zero-mark-up drug policy and reduced prices for diagnostic tests. The purpose was to solve the problem that has plagued China for a long time: it is expensive to visit a doctor. The 2015 THDS is trying to solve the problem of overcrowding in tertiary hospitals.

In the revised manuscript, we have added some explanations regarding the policy on page 4.

b) In the Theoretical Model section (pp 6 and 7) the theoretical model that underlies the statistical analysis seems to be conflated with the statistical model and estimation technique. Provide more justification and explanation why what seem to be three different things can be described as one 'hybrid model', or distinguish them as different components of the analysis.

Response:

Thank you for your rigorous scientific commands. After careful consideration, we conflated the technology of machine learning and the theoretical model. Hence, in the revised main text, we deleted the machine learning from the 'hybrid theoretical framework'. The 'theoretical framework' included Andersen’s behavioral model of health services use and prospect theory. Please see this change in the main text on the page 6-7 in the revised manuscript. We also included the machine learning part separately in the introduction of decision tree analysis and a more detailed explanation in the supplemental file. Please see this change on the page 11 in the revised manuscript.

b)I think the relationship between the Patient Characteristics and Questionnaire Design sections is confusing and needs some reorganization or additional language to clarify

Response:

c)Thank you for your valuable comments. We have moved the section describing the demographic features to the Survey and Questionnaire Design section. Please see it on the page 8-9 in the revised manuscript.

d)Table 1 and its discussion should be moved the the Results section.

Response:

Thank you for your suggestions, we have moved the Table 1 and its discussion to the 

Results section. Please see it on the page 14-15 in the revised manuscript.

e)Reference 27 at the bottom of page 12 (Taeho Jo, Machine Learning Foundations) describing the C4.5 algorithm isn't useful for readers not expert in machine learning (I can't find the term 'C4.5' in the text. The text does clearly use C4.5 or a closely related algorithm, but the general reader won't know this). I have suggested an alternative reference for C4.5, (and also the SVM and MLP algorithms and their relationships to decision trees), or clarify that the reference describes how that type of algorithm works, but don't suggest that the reference mentions C4.5 by name)

Response:

Thanks for your suggestions, and we have replaced the reference as you suggested: Moreno-Ibarra, Villuendas-Rey, Lytra et al. Classification of Diseases Using Machine Learning Algorithms: A Comparative Study. A Comparative Study. Mathematics 2021, 9, 1817. https://doi.org/10.3390/math9151817. We have added the introduction of C4.5 algorithm according to the reference: This is a decision tree type classification algorithm, and this type of classifier is among the most commonly used in classifying patterns, please see it on the page 11 in the revised manuscript; And added ‘the two classifiers were compared by the sensitivity or specificity to the predictive performance in machine learning’, please see it on the page 12 in the revised manuscript.

d) I think the description on pp 13-14 of  the logistic regression analysis says that it is used to validate the decision tree estimates, but also to compare and contrast (as in a sensitivity or robustness analysis?) the results of the two methods. I think this  is confusing and frankly can't be right. I think you are using logistic regression for two distinct functions that it cannot perform simultaneously. I would like clarification of how the logistic regression analysis is used to strengthen the decision tree analysis. I think its use should be characterized as a sensitivity or robustness analysis. Please see my comments in the attachment for more details.

Response:

Thank you for your valuable comments. As per your recommendation, we have changed this statement in our main text: Decision tree and logistic regression are two different classification methods. In many studies, with the same datasets, the two classifiers were compared by the sensitivity or specificity to the predictive performance in machine learning. Lee Yoonju got the common major predictors of suicide attempts in the two methods, showing the robustness of results using the two methods. Hence, we did the logistic regression analysis to compare and advance the results of the decision tree, with the datasets of questionnaires. Please see it on the page 12 in the revised manuscript. 

e) The Analysis of the data from the Population Health Platform on pp 14-15 needs clarification. A clearer explanation of how it relates to the decision tree analysis is needed at the beginning of the section. I think how the Health Platform data is used to confirm the decision tree and logistic regression analysis becomes clear by the end of the presentation of the results later in the paper, but this should be made clear in the Materials and Methods section.

Response:

Thank you for your valuable comments. As per your recommendation, we have reinforced the explanation of the relationship between the two analyses and the necessity of Health Platform data analysis: The previous two analyses used the questionnaire survey data were based on the decision-making of patients' behavior. The survey was limited by the possible inconsistency of patients' external expression and behavioral consciousness and its scope in the city was representative but not comprehensive. Moreover, if different data and methods get similar results, it shows that they are robust. Hence, to verify our conclusions from the questionaries were reliable, we further analyzed additional data available through Jiaxing’s population health platform of outpatients who had visited the city’s hospitals. Please see it on the page 13 in the revised manuscript.

---

## [Editor Report · Decision Letter 2]

24 May 2022

PONE-D-21-20402R2

Relationship  Between O utpatients ’ Sociodemographic and B elief  C haracteristics  and their H ealthcare-seeking  B ehavioral  D ecision-making : E vidence  from  Jiaxing city , China

PLOS ONE

Dear Dr. Yu,

Thank you for submitting your manuscript to PLOS ONE. After careful consideration, we feel that it has merit but does not fully meet PLOS ONE’s publication criteria as it currently stands. Therefore, we invite you to submit a revised version of the manuscript that addresses the points raised during the review process. 

Thank you for your careful and diligent work on the revision. I think the manuscript is almost ready for acceptance. There are two substantive issues preventing acceptance. There are other issues, but these are style and production issues that I think will be easier to handle at acceptance, before the article goes into production.

Substantive issues: 

1) Page 11: Statistical package implementation used to estimate the decision tree. PyCharm is referenced as the 'modeling procedure' (though I think the 'implementation of the decision tree analysis' is better). but PyCharm is a Python programming interface, not an estimation algorithm.  I think the most common implementation of the decision tree algorithm in Python is scikit-learn in the scikit package. However, I know of several custom implementations available on the internet that can be downloaded. I think the official term for things like sci-kit in Python is 'plug-in'. So, the way the tree must have been estimated was to instruct Python to run a specific decision tree estimator plug-in through the PyCharm interface. Please reference the specific plug-in  or source of the code, used to estimate the tree. If I am wrong, please give a reference that shows how you can estimate a tree tree using PyCharm by itself.

2) Page 21: In the Discussion section at the top of the page, you say "Economic factors were the most important for outpatients’ healthcare-seeking behavioral decision-making in China, despite having or not having medical insurance  or belonging to a specific profession." This statement seems to contradict the decision tree analysis where the first decision branch, or split, is whether the subject has health insurance or not. Could you please clarify? Is the discussion at the top of page 21 about more general population behavior in China for seeking any kind of health care? If so what is the source for that information? And how does that seemingly inconsistent piece of information relate the conclusion of the paper, why do you mention it?

Style and production issues

1) Some of the references mix up first and last names of the authors. For example, reference 20 in the main text, the author Amos Tversky is listed as 'Amos T'. The author should be referred to as 'Tversky A'.  Please make sure all the references list the author name property.

2) Please do another check for typographical errors in the text. For example, at the top of page 11 'maching learning' appears instead of 'machine learning'.

3) Please review the text for consistent terminology, especially in the Discussion. For example, at the top of page 20, you say  "In contrast, farmers and industrial workers are more inclined to avoid seeking the highest standard of care." The contrast is with other types of people who seek care at large hospitals. Does 'highest standard of care' refer to 'large hospitals"?  And does 'large hospitals' refer to 'tertiary hospitals"? If so, exactly the same term should be used, otherwise please clarify the contrast. One of the main points of the paper is that large, tertiary hospitals are not really the 'highest standard of care' for minor conditions because other types of hospitals can provide equivalent standard of care for many health problems. I think a review of manuscript is needed to ensure that a clear standard terminology is used to refer to the most important terms used. This will prevent confusion, especially for readers who are not familiar with the Chinese health care system.

4) All the programming languages and specific routines need to be referenced. The references can be very simple. For Stata, the reference I use is  'StataCorp (2017). Stata version 15. College Station, Texas.'

5) The figure for the decision tree is at the end of the main text. For production the style guide says to indicate where in the text you want the figure placed, with the title and description of the Figure. So you should place the contents of the supplemental file 'The annotation of Figure 1' where you want the figure to appear.

6) Add a variable key under Table 2 that explains the meaning of the variables, for example, of 'Prof-3'

7) Thank you for preparing the supplemental file. I think the way you reference the different sections of the supplemental file in the main text is nonstandard and will confuse readers. I did check the style guidelines and the journal is not clear about what system is used to reference supplemental material. I recommend that you change how you reference the supplemental file in the main text to avoid last minute problems in production. For example, at the bottom of page 7, where you have a superscript 'i' I recommend that you simply replace that with the text '(See section S1 of the Supplemental File for more details) ', in normal font (not superscript). Note that the journal requires that  the number of any section, or table or figure in supplemental files must be preceded by an S).

8) in the the Figure of the decision tree, I don't see where the definitions of the three target classifications (A, B, C) are defined. I believe that 'C' refers to tertiary hospitals but I'm not sure. I recommend that A, B, and C be defined in the main text, and clearly related to the terms you use for each health facility, and include that information in the Annotation for Figure 1. Also, the journal if vague on numbering of Figures. When there is only one figure, I can't find in the style guide whether it should be referred to as "Figure' or 'Figure 1'. In any case, if you decide to refer l as 'Figure 1' then make sure that all references to that figure is consistent, including the name of the  file you upload.

If applicable, we recommend that you deposit your laboratory protocols in protocols.io to enhance the reproducibility of your results. Protocols.io assigns your protocol its own identifier (DOI) so that it can be cited independently in the future. For instructions see: https://journals.plos.org/plosone/s/submission-guidelines#loc-laboratory-protocols. Additionally, PLOS ONE offers an option for publishing peer-reviewed Lab Protocol articles, which describe protocols hosted on protocols.io. Read more information on sharing protocols at https://plos.org/protocols?utm_medium=editorial-emailutm_source=authorlettersutm_campaign=protocols.

We look forward to receiving your revised manuscript.

Kind regards,

James M. Lightwood

Academic Editor

PLOS ONE
---

## [Author Response · Author response to Decision Letter 2]

2 Jun 2022

Thank Editor. James M. Lightwood very much for the professional and patient modification, which will bring a lot of benefits to my future research. I am deeply impressed by his(her) rigorous scientific attitude and professional level.

In addition, I hope the journal could speed up the whole progress of decision and could hear from you quilkly.

Thanks again!

---

## [Editor Report · Decision Letter 3]

9 Jun 2022

Relationship Between Outpatients’ Sociodemographic and Belief Characteristics and their Healthcare-seeking Behavioral Decision-making: Evidence from Jiaxing city, China

PONE-D-21-20402R3

Dear Dr. Yu,

We’re pleased to inform you that your manuscript has been judged scientifically suitable for publication and will be formally accepted for publication once it meets all outstanding technical requirements.

There are still some issues with the fine points of academic English style and some typos. I suggest you very carefully review the manuscript when you proof read and edit the final version for publication. For example on page 11 'C4.5 classifier about decision tree model' would be better expressed as 'C4.5 classifier to estimate a decision tree model' On page 24 'effective analysis of the entire population data level' would be better expressed as 'adequate analysis of at the population level'. You'll have an opportunity to do a final proof reading of the manuscript before production. Perhaps you could work with someone fluent in academic English in the final proof reading. I think there are up to a dozen places in the manuscript where English style could be improved. It is unfortunate, but many readers will judge the quality of the research by fine points of formal English style. Improvements in academic English style in a few places in the manuscript, in addition to correction of typos, should be acceptable as long as you don't make any substantive changes to the manuscrpt.

Thank you for your patience with the overly lengthy editorial process. I apologize for the delays.

Kind regards,

James M. Lightwood

Academic Editor

PLOS ONE

---

## [Editor Report · Acceptance letter]

22 Jun 2022

PONE-D-21-20402R3 

Relationship Between Outpatients’ Sociodemographic and Belief Characteristics and their Healthcare-seeking Behavioral Decision-making: Evidence from Jiaxing city, China 

Dear Dr. Yu:

I'm pleased to inform you that your manuscript has been deemed suitable for publication in PLOS ONE. Congratulations! Your manuscript is now with our production department. 

Kind regards, 

on behalf of

Professor James M. Lightwood 

Academic Editor

PLOS ONE